# Hair Transplantation Surgery Versus Other Modalities of Treatment in Androgenetic Alopecia: A Narrative Review

**Swathi Shivakumar** [1], **Martin Kassir** [2], **Lidia Rudnicka** [3], **Hassan Galadari** [4], **Stephan Grabbe** [5] **and Mohamad Goldust** [5,*]

1   Consultant Dermatologist, Deepak Hospital, Bangalore 560070, India; drswathishivakumar@gmail.com
2   Worldwide Laser Institute, Dallas, TX 75231, USA; theskindoctor@aol.com
3   Department of Dermatology, Medical University of Warsaw, 02-091 Warsaw, Poland; lidiarudnicka@gmail.com
4   College of Medicine and Health Sciences, United Arab Emirates University, Al Ain 15551, United Arab Emirates; hgaladari@uaeu.ac.ae
5   Department of Dermatology, University Medical Center of the Johannes Gutenberg University, 55131 Mainz, Germany; stephan.grabbe@unimedizin-mainz.de
*   Correspondence: mgoldust@uni-mainz.de

**Abstract:** Androgenetic alopecia (AGA) is the most common type of baldness and its incidence has increased over the past few years with an earlier age of onset being widely reported all over the world. Although it is reported more often in men, it affects women as well. With the growing cosmetic concern of patients, emphasis has shifted from the more traditional treatment options such as finasteride and minoxidil to surgical options such as hair transplantation. This review briefly highlights all of the treatment options available for AGA so far. A special focus is on current data available on hair transplantation surgeries and the various methods, merits and demerits and limitations of surgery. The literature research considered published journal articles (scientific reviews) from 1990 to date. Studies were identified by searching electronic databases (MEDLINE and PubMed) and the reference lists of respective articles. Only articles available in English were considered for this review.

**Keywords:** androgenetic alopecia; treatment; hair transplantation; follicular unit extraction



## 1. Introduction

Androgenetic Alopecia (AGA) is a genetic type of hair loss with varying incidence and severity across different age groups and races [1]. The androgen testosterone in its active form i.e., dihydrotestosterone (DHT) acts on receptors on the hair follicle and causes its gradual miniaturization. This converts terminal hair into vellus hair, eventually leading to baldness [2].

As these androgen receptors are expressed to a greater degree in the hair follicles of the frontal and vertex region of the scalp in men, AGA manifests as a progressive pattern type of hair loss affecting these regions with a relative sparing of the occipital and temporal regions [3,4]. The grading system followed for male AGA is the Norwood–Hamilton scale [1]. In women, there is diffuse thinning mainly at the crown with a widening of the hair partition and this is graded using various scales such as the Sinclair, Ludwig, Ebling and Olsen scales [5].

Treatment is aimed at reversing this androgen mediated hair miniaturization process as well as providing cosmetic coverage of the bald area. While there are multiple options available, there is no single solution for all. It has to be tailored according to the needs of the patients. The best results are obtained when a combination of treatments are used.

## 2. Materials and Methods

The literature research considered published journal articles (clinical trials or scientific reviews). Studies were identified by searching electronic databases (MEDLINE and PubMed) and the reference lists of respective articles. Only articles available in English were considered for this review.

## 3. Discussion

### 3.1. Treatment Options

### 3.1.1. 5-Alfa-Reductase Inhibitors

5-alfa-reductase is the enzyme responsible for the conversion of testosterone to its active form of dihydrotestosterone (DHT). DHT binds to its receptors on hair follicles and leads to its gradual miniaturization [6]. Therefore, the drugs targeting this enzyme are used for the treatment of AGA. Two drugs belonging to this class have been studied, finasteride and dutasteride.

Finasteride inhibits the type-II 5-alfa-reductase enzyme [6]. Topical formulations are also available as a combination with monoxide [7]. Oral finasteride has been approved by the US-FDA for the treatment of AGA [6].

Dutasteride acts by inhibiting both type-I as well as type-II isoenzymes of 5-alfa-reductase [8].

In males, a dose of 1 mg of finasteride and dutasteride at a dose of 0.5 mg are used.

The dosage used for females is variable in different studies. In a randomized controlled trial evaluating its efficacy in females, finasteride at a dose of 1.25 mg and dutasteride at a dose of 0.15 mg was given to two groups and the patients were evaluated over a three-year period. A statistically significant improvement was observed in both groups. Dutasteride produced better results in the younger age group i.e., <50 years and in the central vertex sites [9].

Studies have shown statistically significant results in the improvement of hair thickness and the rate of hair growth as well as an increase in the duration of the anagen/growth phase [10].

An RCT comparing the efficacy of dutasteride with finasteride showed statistically significant better hair growth as well as a reversal of miniaturization in the dutasteride group compared with the finasteride group [11].

Side effects, although less with the low dose used, are often troublesome and unacceptable to most patients. These include low libido, erectile dysfunction and other sexual side effects due to the reduction in the circulating levels of DHT [12]. They are reversible on discontinuation of the drug but in a few cases may persist for ≥3 months [13].

### 3.1.2. Minoxidil

Other than finasteride, minoxidil is the only other drug that is approved by the US-FDA for the treatment of AGA [14] After conversion to its active form, i.e., minoxidil sulfate, it acts on the receptors on arterial smooth muscle cells to cause vasodilatation. This leads to a prolonged anagen of the miniaturized hair follicle and reduced shedding [15].

Minoxidil is available as 2%, 5% and 10% strength. The recommended dose is 1 mL twice daily [16]. The side effects are minimal and restricted to the treatment site. This includes itching, scaling and hypertrichosis of the forehead due to dripping. This can be reduced by using the alcohol-free foam preparation. The major limiting factor is the need for the continued use of the drug as the positive effects wear out once the drug is discontinued [17].

Recently, there have been various reports of the use of minoxidil in its oral form in different dosages. One study reported its use at a dose of 0.25–1.25 mg for female pattern hair loss and 2.5–5 mg for male pattern hair loss. The main reason for the shift to the oral form was due to the compliance issue associated with the topical form. However, it caused side effects in 29.3% of the study group with hypertrichosis being the most common (24%) followed by pedal edema (4.8%) [18].

### 3.1.3. Platelet Rich Plasma Therapy

Activated platelets are known to release growth factors and cytokines (PDGF) and hence are used for the treatment of various conditions such as stimulating wound healing and hair growth. Its popularity has increased in recent years and much research has been done to evaluate its efficacy in AGA. The growth factors released from activated platelets were found to stimulate the stem cells in the hair bulge leading to folliculogenesis, neovascularization and an increased duration of the anagen phase.

A total of 10 mL of the blood of a patient is withdrawn and centrifuged to separate the plasma concentrate. This is then loaded into insulin syringes and injected into the scalp. At least 4–6 sessions are recommended, 2–3 weeks apart. The most common side effect noted was pain at the site of the injection. A lack of standardization of the platelet rich plasma process is a major factor leading to conflicting results of previous studies regarding its efficacy [17].

### 3.1.4. Microneedling

The process of creating controlled tissue injury and microchannels using devices such as dermarollers/electric pens is known as microneedling. These devices contain multiple needles of varying diameters and lengths and are selected according to the purpose of their use. They stimulate hair growth by the release of growth factors such as the platelet derived growth factor (PDGF), which activates stem cells in the hair bulge [19].

### 3.1.5. Stem Cell Therapy

Stem cell therapy possibly represents the future of the management of all ailments including pattern hair loss. However, data are very limited due to the ethical considerations as well as the probable high cost involving the extraction.

Stem cells can be derived from either the bone marrow, adipocytes or the hair follicle bulge.

Follicle stem cells are obtained by a 4 mm punch biopsy from the scalp, which is processed and injected into the bald areas [20].

Fat cells/adipocytes are a rich source of stem cells that can be utilized for the stimulation of hair growth. The process involves the harvesting of a superficial layer of fat from the abdomen by liposuction and injecting it into the balding areas of the scalp after processing.

Although it is potentially a very promising therapy option in the future, further research and larger clinical trials are required before stem cell therapy can enter into mainstream medicine [21].

### 3.1.6. Electrotrichogenesis

Electrotrichogenesis is a process of the application of a pulsed electric field to the scalp to stimulate hair growth. Maddin et al. performed a study in which 73 male patients with AGA grade III and Grade IV were treated with electrotrichogenesis over 36 weeks. At the end of the study, there was a significant increase in the hair count in the treatment group [22].

### 3.1.7. Fractional Radiofrequency

Fractional radiofrequency (Rf) as a treatment option for AGA was studied by Verner and Lotti using HairLux (Innogen Technology Ltd., Yokneem, Israel). Twenty-five patients were enrolled into the study and they received ten treatments, two weeks apart. There was an improvement in the hair density and thickness. The treatment was found to be well tolerated [23].

### 3.1.8. Low Level Laser Light Therapy (LLLT)

Lasers in low doses have been utilized in the treatment of hair loss including AGA. The probable mechanism of action is through the generation of anti-inflammatory cytokines and anti-oxidants, which stimulate hair growth. Studies done up to the present date have

reported a significant improvement in hair density and diameter in patients treated with LLLT [24–26].

### 3.1.9. Biofibre Hair Implant

Biofibre® medical hair prosthetic fibers have all of the biocompatibility and safety requirements established by international standards for medical devices. Tchernev and his colleagues demonstrated that Biofibre® hair implants are safe and well tolerated by patients and can be totally reversible if the need arises [24].

### 3.1.10. Hair Transplantation

Hair transplantation (HT) is a method of transferring hair follicles from a donor area to a recipient/bald area. The gold standard for hair transplant is the follicular unit transplant (FUT). The most commonly chosen donor site is the occipital region as the hair in this region is unaffected by AGA. When the hair in the occipital region is scanty or insufficient, facial hair and body hair may be used. The rationale for hair transplantation in AGA is donor dominance, i.e., the transplanted hair continues to be unaffected by the balding process even after relocation to the new site [25].

Pre-operative laboratory investigations include hemoglobin, total count, platelets, bleeding and clotting time, prothrombin time, activated partial thromboplastin time, blood sugar and viral markers, i.e., HIV and Hep B surface antigens. A chest X-ray, electrocardiogram and physician fitness for surgery are also obtained prior to the surgery. If a patient is using minoxidil, it is stopped two weeks prior to the procedure. The patient is also advised to avoid smoking and taking NSAIDs for at least one week before the surgery [25].

The two methods of donor harvesting in a hair transplant are the traditional strip method and the more recent technique known as the follicular unit extraction (FUE). The strip harvesting technique has the drawback of leaving a cosmetically unacceptable linear scar. FUE utilizes small circular punches to extract individual follicular units. There are both manual and mechanical punches available. Although mechanical punches considerably reduce the operation duration, they also increase the chance of a graft transection. Individual tailoring of the punch size and depth depending on the patient is key to minimizing the transection [26].

Although hair transplantation is a simple procedure done under local anesthesia, there are a few absolute contraindications for the procedure such as an active autoimmune disease, uncontrolled diabetes/hypertension and patients allergic to local anesthetic, i.e., lignocaine. Patients younger than 25 years old tend to have unrealistic expectations and have progressive hair loss therefore they should be discouraged from undertaking HT [27].

For the traditional strip method, a preoperative assessment of the occipital scalp laxity is essential to prevent healing with excessive wound tension. It is also essential to note the density so as to determine the number of grafts that can be harvested in one session. Characteristics of the grafted hair also affect the results of transplant. Course/curly hair appears denser than fine, straight hair [28]. The scar of the strip surgery can be repaired by grafts extracted by FUE from either scalp/body hair or camouflaged by micropigmentation [29].

Harvesting grafts as follicular units from the donor area was first described by Woods and hence is named the Woods Technique. It was renamed as the follicular unit extraction (FUE) by Rassman et al. in 2002 [30]. While planning to perform an FUE, it is important to note the angle of hair growth to minimize transection. The number of grafts should not exceed 35% of donor hair density in the first sitting and 10–20% in the second sitting [29].

In the preoperative planning, the scalp hair is trimmed to 3 mm in length. A few grafts can be obtained by FUE from the occipital area and examined under a stereomicroscope to assess their integrity using the Fox Test. According to this system, class 4 and 5 are considered as Fox negative and are unsuitable candidates for transplantation [29].

The anesthetic used most commonly is 2% lignocaine with adrenaline at a dose not exceeding 6–7 mg/kg of the body weight. In addition, a tumescent saline solution with epinephrine can be injected, which produces vasoconstriction and reduces intra-

operative bleeding. It also lifts the sub-cutis from the underlying vessels and nerves thereby preventing its damage [25].

With a graft extraction, the punch is held with the dominant hand and the epidermis and dermis is dissected down to the follicle paying special attention to the angulation of the hair shaft while the non-dominant hand applies counter-pressure. The neighboring 2–3 hair follicles are skipped and the next one is dissected using the same technique. This is repeated until the required number of grafts is obtained. The extracted grafts are placed in sterile normal saline and later dissected and placed over a wet saline-soaked gauze. These units are now ready for placement into the recipient area [29].

With a graft placement, the recipient area is anesthetized with lignocaine and adrenaline injections. Serial punctures are made with a needle in accordance with the design of the anterior hairline and the angulation of the hair shaft. The grafts are then placed into these sites [29].

HT is a day care surgery and the patient is usually discharged on the same day. Most surgeons do not prefer to bandage the recipient site so as to avoid the grafts from sticking to the bandage and getting dislodged. Swelling is a very common sequelae especially in an older age group and can be minimized by the use of a pressure bandage on the forehead. A few authors have also reported using an injection of triamcinolone during the procedure or oral steroids for three days post op to minimize edema.

It takes at least 6–9 months for the full results of the hair transplant to be visible. In the case of larger areas of balding needing to be covered, it should be planned in multiple sittings with a gap of at least 4–6 months after the previous treatment [31].

The most common complications of hair restoration surgery are usually mild and self-limiting. The most commonly observed complications are bleeding, infection and pain. If the procedure is done under strict aseptic conditions the chances of infection are minimal. It can be managed with topical anti-bacterial creams and analgesics if required [30].

The common reasons for dissatisfaction in a patient include a poorly designed hairline and a lack of counseling regarding the progressive nature of the hair loss. The hairline needs to be designed by an experienced dermatosurgeon prior to the surgery keeping in mind the temporal angles and making sure that they are bilaterally symmetrical. The distance between the frontal hairline and the eyebrows also needs to be planned as a very low-lying hairline can give a very unnatural appearance [32].

## 4. Conclusions

Although this review focuses on hair transplantation, it is important to note that there is no single treatment modality for male pattern hair loss that is 100% effective. To date, oral finasteride and topical minoxidil continue to be the only US-FDA approved medications for AGA. Hair transplantation, while providing good cosmetic coverage for the bald areas, does not halt the progression of the disease. Hence, medications such as minoxidil and finasteride will need to be continued even after surgery. Management requires a multi-disciplinary approach, which has to be individualized keeping in mind the expectations of patients, affordability, age and grade of hair loss.

*Limitations of This Study*

Only articles available in English were reviewed in this study.

**Author Contributions:** Conceptualization: S.S., L.R., M.G.; Methodology: S.S., M.K., L.R., H.G., S.G.; Software: H.G., S.G.; Validation: M.K., L.R., H.G., S.G.; formal analysis: L.R., H.G.; investigation: S.G., M.G.; resources: S.S., M.K., L.R.; data curation: L.R., H.G., S.G.; writing—original draft preparation: S.S., M.G.; writing—review and editing: S.S., M.K., L.R., H.G., S.G.; visualization: S.S., M.G.; supervision: M.G.; project administration: S.S., M.K., L.R., H.G., S.G., M.G. All authors have read and agreed to the published version of the manuscript.

**Funding:** This research received no external funding.

**Institutional Review Board Statement:** Not applicable.

**Informed Consent Statement:** Not applicable.

**Conflicts of Interest:** The authors declare no conflict of interest.

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
