# Peer review of "Hair Transplantation Surgery Versus Other Modalities of Treatment in Androgenetic Alopecia: A Narrative Review"

_cosmetics, doi:10.3390/cosmetics8010025_

Round 1

Reviewer 1 Report

Authors present a narrative review about the role of hair transplantation surgery in patients with AGA. This is an interesting topic, the main limitation is that the review is very superficial and does not separate the information regarding men and women with AGA (results, side effects, surgery...)

1.- Authors should define the objetive of the review as they include medical therapies and surgery

2.-Please define better the search strategy (time period, type of studies...) Authors indicate that they have reviewed clinical trials but they do not present any clinical results (efficacy, side effects...)

3.- Please consider, in the introduction, differentiate between the occipital area and the vertex

4.- Authors mention some grading system, but there are many scales (Ebling, Ludwig, Olsen...)

5.- Please indicate differences between finasteride and dutasteride prescription (dose, results...) in men and women and the results of clinical trials. More information regarding side effects should be included

6.-Recently many papers have described the prescription of oral minoxidil in AGA. Please add a comment. 

7.-Stem cell therapy is under investigation nowadays. There is not any advanced therapy approved for AGA. Please review this paper (PMID: 33182308)

8.- Please include more information about clinical results after surgery, side effects, and if patients should continue medical therapy after surgery.

Author Response

Response to reviewer-1

Thank you for your valuable suggestions. Below is a point by point response to the corrections pointed out-

1.- Authors should define the objective of the review as they include medical therapies and surgery

Objective of the review is to highlight the role of hair transplant surgery and compare it to other modalities of treatment available so far for AGA. In compliance with the same, I have changed the title- Hair Transplantation Surgery vs Other Modalities of Treatment in Androgenetic Alopecia: A Narrative Review

2.-Please define better the search strategy (time period, type of studies...) Authors indicate that they have reviewed clinical trials but they do not present any clinical results (efficacy, side effects...

Time period- 1990 till date, review articles as well as clinical trials. Clinical results of studies on 5-alfa reductase inhibitors as well as oral minoxidil have been added- page 2,line 62-75; page 2 line 87-92.

3.- Please consider, in the introduction, differentiate between the occipital area and the vertex.

Page 1, Line 33

4.- Authors mention some grading system, but there are many scales (Ebling, Ludwig, Olsen...)

Page 1, line 37

5.- Please indicate differences between finasteride and dutasteride prescription (dose, results...) in men and women and the results of clinical trials. More information regarding side effects should be included

Page 2, line 56-75

6.-Recently many papers have described the prescription of oral minoxidil in AGA. Please add a comment. 

Page 2, line 87-92

7.-Stem cell therapy is under investigation nowadays. There is not any advanced therapy approved for AGA. Please review this paper (PMID: 33182308)

Page 3, line 114-120

8.- Please include more information about clinical results after surgery, side effects, and if patients should continue medical therapy after surgery.

Page 5, line 208-217; Page 5, line 208-216; Page 5, line 224-225

Reviewer 2 Report

This is a bibliographical review. The review considered published journal articles by searching electronic databases (MEDLINE and PubMed). Only articles in English have been taken into account. This is a first limitation.  

This review briefly highlights all the treatment options available for Androgenetic alopecia (AGA), including a section (3.1. Treatment options) with 10 different subsections (3.1.1 to 3.1.10). However,  according the title or according to the extension of the different option, only the hair transplantation surgical option is really revised. The various methods, merits and demerits, and limitations of surgery are certainly discussed. The first 9 options are too briefly and treated in a very superficial way.  Authors should decide what they want to review. General treatments of AGA or hair transplantation surgery approach. But the current version has not the standard quality for publication in cosmetics.

For instance, section 3.2 has incomplete information with undefined units (% of strength is an undefined unit, and 1 ml is not related to that. The differences between finasteride and dutasteride and the reason the amount is different are not justified, and commercial names should be omitted (Proscar, Avodart). PDGF should be mentioned in the section 3.1.3 rather than 3.1.4 and then the link among needles and PDGF should be discussed. Processing of stem cell therapy or electrotrichogenesis are not described and are not discussed. A single reference published 30 years ago is poor.

In summary, the topic is of course interesting and within the field of interest of Cosmetics, but authors should decide a review about AGA for the overall point of view, o a minireview on hair transplantation surgical. In the last option, some order and re-writing of this section is also needed. Conclusion should be coherent and consonant with the content of the review.

Format should be also repaired and corrected… i.e abstract spacing. References should have an uniform style

Author Response

Response to reviewer-2

Thank you for your valuable suggestions. Kindly find the changes incorporated.

This is a bibliographical review. The review considered published journal articles by searching electronic databases (MEDLINE and PubMed). Only articles in English have been taken into account. This is a first limitation.

This is indeed a limitation as pointed out, and will be mentioned at the end of the study. Page 5, line 241.

This review briefly highlights all the treatment options available for Androgenetic alopecia (AGA), including a section (3.1. Treatment options) with 10 different subsections (3.1.1 to 3.1.10). However,  according the title or according to the extension of the different option, only the hair transplantation surgical option is really revised. The various methods, merits and demerits, and limitations of surgery are certainly discussed. The first 9 options are too briefly and treated in a very superficial way.  Authors should decide what they want to review. General treatments of AGA or hair transplantation surgery approach. But the current version has not the standard quality for publication in cosmetics.

The title has been revised accordingly, to show the main aim of the study. Also , various clinical trial reports have been added to discuss 5alfa reductase inhibitors, oral minoxidil and stem cell therapy.

For instance, section 3.2 has incomplete information with undefined units (% of strength is an undefined unit, and 1 ml is not related to that. The differences between finasteride and dutasteride and the reason the amount is different are not justified, and commercial names should be omitted (Proscar, Avodart).

Page 2, line 83-93.

PDGF should be mentioned in the section 3.1.3 rather than 3.1.4 and then the link among needles and PDGF should be discussed.

Page 3, line 96.

 Processing of stem cell therapy or electrotrichogenesis are not described and are not discussed. A single reference published 30 years ago is poor.

Page 3, line118-125

In summary, the topic is of course interesting and within the field of interest of Cosmetics, but authors should decide a review about AGA for the overall point of view, o a minireview on hair transplantation surgical. In the last option, some order and re-writing of this section is also needed. Conclusion should be coherent and consonant with the content of the review.

Format should be also repaired and corrected… i.e abstract spacing. References should have an uniform style

Round 2

Reviewer 2 Report

The manuscript has been improved and most of the points raised by reviewers have been addressed. The title has been changed, and this one is in agreement to the review focusing. Other changes have also convenient, although the review is still a little bit poor in some sections aside the main one concerning hair transplantation surgery. Anyway, the review is now suitable for publication. Before that, a couple of minor changes concerning the format:

  • Section 1.1.5. alfa reductase inhibitors

Please, separate the enzyme 5-alfa-reductase from the numerical classification 3.1.1.

5-alfa-reductase would be linked by a hyphen.

  • References 1 and 10. Please, replace capital letters in the titles of the article.

Author Response

The minor corrections suggested by reviewer 2 have been incorporated.